# The Intensification of Parenting in Germany: The Role of Socioeconomic Background and Family Form

Sabine Walper [1,*] and Michaela Kreyenfeld [2]

1   German Youth Institute (DJI), Nockherstr. 2, 81541 Munich, Germany
2   Hertie School, Friedrichstr. 180, 10117 Berlin, Germany; kreyenfeld@hertie-school.org
*   Correspondence: walper@dji.de

**Abstract:** Drawing on the international discourse on the intensification of parenting and new data from Germany, this paper aims to contribute to a better understanding of the unique challenges that parents face in the 21st century. We used data from the survey "Parenthood Today", which was conducted in 2019 to examine parents' views on parenting in Germany. The data comprised standardized interviews with 1652 mothers and fathers. We focused on three dimensions of parental pressures: namely, time pressure, financial pressure, and pressure that emanates from the educational system. Time pressure referred to the pressure currently felt, whereas financial pressure and pressure from the educational system referred to changes across time. In each of these domains, more than 60% of the parents experienced high (time) or increasing (education and financial) pressure. Binary logistic regressions showed that while parental education was a strong predictor of experiencing an increase in financial pressure, parental education did not matter for other realms of parenting. However, employment and family form were strongly related to parental time pressure. Full-time employed lone mothers, but also non-resident fathers, reported experiencing heavy pressure when trying to balance their roles as a worker and as a carer. Our results draw attention to the importance of better integrating the needs of post-separation families, including of non-resident fathers, in the debate on the "intensification of parenting".

**Keywords:** parenting; intensification of parenting; family diversity; lone mothers; non-resident fathers; socioeconomic background

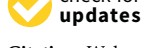



## 1. Introduction

Increasing ratios of non-marital childbearing, union instability, re-partnering after union dissolution, as well as family formation among same-sex couples have contributed to a growing diversity of family forms. Furthermore, migration has led to an increase in ethnic-cultural diversity among families in many countries of Europe. At the same time, globalization and digitalization have transformed labor markets, which have, in turn, generated new social cleavages and inequalities between families. While these trends have been identified by researchers, the media, and policy makers as major social policy concerns, changes in parenting norms and parenting practices have received less attention in these debates. The growing expectations and demands that parents experience in many contemporary societies have recently been captured under the theme of the "intensification of parenting" (e.g., Faircloth 2014; Smyth and Craig 2017).

Family researchers in the U.S. and Great Britain have been the first to address this trend toward parents putting increasing effort into the care and upbringing of their children (e.g., Bianchi 2000; Bianchi et al. 2004; Craig et al. 2014; Faircloth 2014; Vincent and Maxwell 2016). In both countries, factors like the dominance of private child care and private education and growing inequality at the societal level have driven this trend. However, in continental Europe—including in Germany, which will be the focus of our analyses —conditions differ. Educational systems are primarily public (for details, see below). Furthermore,

changes in social inequality at the societal level have been less extreme in Germany, where family policies aim to reduce socioeconomic differences across families, and to alleviate some of the challenges associated with reconciling work and child care (BMFSFJ 2021). Nevertheless, concerns have also been raised for Germany that parenting has become increasingly demanding and that socioeconomic differences in parents' abilities to meet these demands may generate new forms of social inequality (e.g., Henry-Huthmacher et al. 2008; Ruckdeschel 2015).

The intensification of parenting manifests in an increase in the material and, above all, the immaterial investments of parents in the upbringing, education, and care of their children, which include devoting more quality time to children and making more concerted efforts to support and foster children's positive development. The standards of what constitutes "good parenting" have risen with the emergence of the burgeoning literature on how targeted parental input can promote child development and maximize children's acquisition of competencies (Wall 2010). Particularly in societies that emphasize parents' individual responsibility for their children's upbringing, rising standards seem to have fueled a competition in parental investments in childrearing (Faircloth 2020). As high-income parents have been better able than low-income parents to respond to these new demands, this trend has further exacerbated social disparities in parenting styles, and in parental investments of time and money in children. Overall, parental behavior can be seen as a new dimension that has enforced social inequalities (e.g., Dotti Sani and Treas 2016; Vincent and Maxwell 2016).

Concerns about the effects of the intensification of parenting were addressed in the Ninth Family Report for Germany, which was published by the German Federal Family Ministry in 2021 (BMFSFJ 2021). Based on the report of an interdisciplinary expert committee and framed by a detailed statement of the government, this family report provides deep insights into the demographic, social, and economic situations of families, and into parenting behavior in Germany. It describes trends, illustrates relevant legal conditions, and provides concrete policy suggestions. As part of our work in the expert committee, we not only collected the available evidence from official statistics and compiled the available empirical evidence from family research; we also launched our own survey that focused on "Parenthood Today" (Institut für Demoskopie Allensbach 2020).

In the following, we address the unique challenges faced by parents in the 21st century. We first review international research on the intensification of parenting. We then contextualize these findings for the German case. The empirical part presents analyses of the survey "Parenting Today". Based on this survey, we provide novel evidence on the intensification of parenting, as indicated by parents' reports of feeling that the pressure of parenting has been increasing. Furthermore, we explore the social disparities related to these pressures for the German case.

## 2. The Intensification of Parenting in International Perspective

### 2.1. Parents' Monetary and Time Investments in Children

The hypothesis of an intensification of parenting is based on findings from several industrialized countries, which indicate that parents are investing more time in child care today than they were in the 1960s (e.g., Bianchi 2000; Dotti Sani and Treas 2016; Gauthier et al. 2004; Gimenez-Nadal and Sevilla 2012). Fathers, but also mothers, have increased their levels of engagement in child care, even though maternal employment rates have risen in many countries in recent years. In the same vein, data on children's time use in the U.S. indicate that the time children are spending with their parents has increased, even as children's participation in preschool and school programs has expanded (Hofferth and Sandberg 2001). While parents in some countries may not be devoting more time to child care, the time that parents spend with their children is more likely to be "exclusive time". For example, data from Australia for the period between 1992 and 2006 showed that parents reported engaging in more child-centered activities, even though

the overall time they were spending with their children was declining (Craig et al. 2014). Findings along these lines appear to be pointing to a compression of "quality time".

Of particular social policy relevance is the question of whether the trend toward intensified parenting is uniform across social strata, or whether it differs by socioeconomic background and parental income. Across countries, higher educated parents spend more time with their children than parents with lower educational resources (Guryan et al. 2008). However, the abovementioned study for Australia reported a decline in disparities in the time spent with children between different educational groups from 1992 to 2006 (Craig et al. 2014). In an analysis of U.S. data collected between 1965 and 2000, Bianchi et al. (2004) found no changes in the educational gradient of parents' investments of time and money in their children. However, more recent findings from the U.S. suggest that parents' investments in quality time have become less equally distributed across social strata. A comparison of time use data from 1965 to 2013 that captured active parental involvement in developing their children's social, cognitive, or linguistic skills (developmental care time) revealed that mothers and fathers from all educational groups were spending more time on developmental care, but that this increase was more pronounced among mothers with a bachelor's degree (Altintas 2016). Thus, it appears that the positive educational gradient in the amount of active quality time spent on child care has become steeper. Similarly, a cross-national study of 11 industrialized countries found that educational disparities increased between 1965 and 2012, with more highly educated parents investing progressively more time in their children (Dotti Sani and Treas 2016). This study also reported that educational disparities increased consistently across all countries during this period, which suggests that disparities in child investments by socioeconomic background were growing.

In line with these trends, children's time use has also changed. In the U.S. between 1981 and 1997, children's participation in structured activities, such as school, time in day care, sports, and artistic activities, increased. Over the same period, the time children spent on less structured activities, such as playing, watching TV, meeting friends, and "passive leisure", decreased (Hofferth 2009; Hofferth and Sandberg 2001). In the subsequent period from 1997 to 2003, children's time spent on less structured, self-determined activities also declined; whereas the time children invested in structured activities, such as involvement in youth organizations, increased. As Hofferth (2009) noted, parents may have been concerned about "overbooking" their children's schedules and thus prioritized education-oriented activities, which were seen as key for children's future opportunities.

An intensification of parenting can also be observed with respect to parents' monetary investments in their children. There is consistent empirical evidence that expenditures on children are strongly correlated with parental income. While Bianchi et al. (2004) showed that this correlation remained stable between 1988 and 1998 in the U.S., other studies covering a wider time range concluded that the association between parental income and investment per child was becoming stronger over time. Research using data from the 1970s to 2010 found that spending on children increased over this period, both in absolute terms as well as in relation to families' household income (Duncan and Murnane 2011; Kornrich and Furstenberg 2013). The findings further indicated that while this increase was evident across all social strata, high-income parents were making larger monetary investments in their children than low-income parents, and that this difference was growing over the years. It has also been shown that social inequalities, measured at the macro level of society, were related to individual investments in children. If social inequality was high at the societal level, the gap in child-related monetary investments between parents of different income strata was also greater (Schneider et al. 2018).

### 2.2. Intensive Parenting Styles

The discussion about the intensification of parenting was backed up not only by findings on parents' time use and monetary investments, but also by changes in parenting behavior. Parents have increasingly come to see themselves as highly responsible for monitoring their children's activities and managing their affairs, even in late adolescence

and early adulthood (Gauthier 2015; Kouros et al. 2017; LeMoyne and Buchanan 2011; Padilla-Walker and Nelson 2012; Schiffrin et al. 2014). In the popular literature, this type of parental behavior is frequently referred to as "helicopter parenting," but research has also characterized it as "hyperparenting" (Janssen 2015) or "overparenting" (Segrin et al. 2013). Concerns have been raised that this type of intensified parental involvement may inhibit child development, as it does not sufficiently take into account the child's developmental stage. These parenting practices may shield children from making their own choices, and can thus prevent them from developing personal responsibility and competence. There is, for example, evidence that children who experience highly controlling parenting during early childhood demonstrate less ability to self-regulate during preadolescence (Perry et al. 2018).

"Helicopter parenting" seems to represent an extreme manifestation of broader changes in parents' understanding of how they should best fulfill their roles. Over the past several decades, the shift away from authoritarian-hierarchical family structures and toward more child-centered parenting behaviors has been well documented (Doepke and Zilibotti 2019; Park et al. 2014; Schneewind and Ruppert 1995). This shift places higher demands on parents to provide a beneficial context for child development, as a child's voice is assigned greater importance, and family rules are increasingly negotiated. It has been argued that compared to authoritarian, neglectful, but also indulgent parenting, authoritative parenting is particularly beneficial for promoting positive child development, as it combines parents being responsive to their child's needs, while also demanding that the child exhibits competent behavior (Baumrind 2013; Steinberg 2001). Thus, authoritative parenting has become the standard for "ideal parenting". Along with this trend, parents' demands for guidance have increased markedly. Faircloth (2014) noted a significant increase in the publication of advice literature and academic books on the care, nurturing, and upbringing of children that started in the late 1960s, accelerated in the second half of the 1970s, and did not level off until around the end of the 1990s.

A number of factors have been identified as the main sources of these changes. Findings from research on attachment, parenting, and education have contributed to an increasing pedagogization of childhood, and especially of early childhood. Due to new discoveries about the developmental dynamics of the first years of life and the role of early interaction experiences and learning opportunities in children's further development, parents are increasingly advised to engage in child-centered, responsive childrearing practices, and provide stimulating educational experiences in these early phases of their child's development (e.g., Nationale Akademie der Wissenschaften Leopoldina and der Wissenschaften 2014; Wall 2010, 2018). Attachment research in particular has emphasized the importance of parental sensitivity to the child's needs as the key to secure attachment (e.g., De Woolff and van IJzendoorn 1997), which has, in turn, been identified as a salient factor in a child's positive development (e.g., Li et al. 2021; Meins 2013; Zimmermann et al. 2001). Attachment research has contributed to a better understanding of the role of parental sensitivity, and has also provided avenues for promoting parenting skills (e.g., Landry et al. 2006; van den Boom 1995). At the same time, however, it has raised the standards of engagement for parents, and especially for mothers.

Much of the discourse on intensive parenting has focused the role of mothers. For example, Liss et al. (2013) argued that the intensification of parenting has primarily affected mothers due to five widespread beliefs, i.e., the belief that (1) mothers are inherently better parents than fathers; (2) childrearing has to be fulfilling; (3) parents are responsible for promoting their children's development; (4) motherhood is challenging; and (5) parents should prioritize their children's needs and over their own needs (Liss et al. 2013). However, the high (self-)attribution of responsibility to mothers appears to be problematic, as mothers who subscribe to these beliefs have been found to report lower life satisfaction and more mental health problems (Rizzo et al. 2013). In addition, it has been suggested that this model of childrearing overloads the role of motherhood and fosters traditional gender roles.

The changes in parenting styles must also be seen in conjunction with the mounting pressure associated with increasing labor market competition. A comparison of countries showed that in countries with greater socioeconomic disparities, parents tend to have higher educational aspirations for their children, face more pressure to spend quality time with their children, and have more intensive styles of parenting (Doepke and Zilibotti 2019). Large social inequalities at the societal level have increased the value of parents' investments in their children, since the "returns" on such efforts rise with corresponding opportunities for advancement (as does the fear of downward mobility if the children fail to meet these standards). According to findings by Doepke and Zilibotti (2019), parenting has become noticeably more intense in neoliberal countries such as the U.S., while in countries such as Sweden, which are characterized by lower social disparities, children are under less educational pressure, and are given more freedom to develop in the direction they choose.

Parents' socioeconomic resources have often been linked to their parenting values and practices, while different theoretical notions have been cited in explaining these links. Social class differences in parenting have been attributed to differences in the economic pressures families face. These pressures are greater among families who have fewer financial resources, and who are at a higher risk of failing to meet the standards of intensive parenting (Conger et al. 2010). Social class differences in parenting have also been shown to reflect differences in parents' job-related experiences and associated expectations about what matters in a child's upbringing (Kohn 1969). Furthermore, differences in cultural models of parenting have been identified that may be linked to the trends outlined above. While higher educated parents tend to closely follow the model of "concerted cultivation" (Lareau 2003), and thus deliberately seek to integrate a wide range of stimulating learning options into their everyday family life, less educated parents are more likely to follow the parenting model of "letting things grow." Thus, in addition to having different daily pressures, lifestyles, and attitudes, parents are likely to differ in their expectations about the returns of intensive childrearing efforts, which may explain some of the observed differences in parenting behavior across socioeconomic groups.

Finally, parenting values and practices may determine fertility choices. If the costs and investments per child are perceived as being too high, couples may decide to remain childless or to have only one child. For example, the lowest-low fertility and increasing childlessness in countries such as Japan or South Korea have been linked to the increasing demands of the educational systems in these countries (e.g., Fleckenstein and Lee 2019).

### 2.3. Summary and Research Question

To conclude, prior research has provided strong evidence on an intensification of parenting along several dimensions, including investments in time, financial investments, and investments in children's education (Bianchi 2000; Craig et al. 2014; Doepke and Zilibotti 2019; Hays 1996; Vincent and Maxwell 2016). The results of the existing studies also suggest that parents' investments in their children have become increasingly unequal (Kouros et al. 2017; LeMoyne and Buchanan 2011; Padilla-Walker and Nelson 2012; Schiffrin et al. 2014). Much of this research was conducted among parents in couple households. Although the time pressures faced by single parents have been addressed in this literature (Bakker and Karsten 2013; Hertz and Ferguson 1998; Kendig and Bianchi 2008), very little attention has been paid to how parental separation and family diversity are related to the intensification of parenthood. A glaring gap in the literature is that non-resident parents have been completely left out of this discourse.

Another gap in the existing literature is that most of the previous studies on the intensification of parenthood were conducted in the Anglo-American context. In these countries, a strong association between societal inequalities and investments in children has been found (e.g., Schneider et al. 2018). However, it is less clear whether these findings translate well to other contexts where social inequality has risen less sharply. In contrast to the Anglo-American context, most welfare states in continental Europe have a public education system that is used by students of all socioeconomic classes. Furthermore,

many European countries have enacted family policies aimed at alleviating some of the time pressures families face, and at reducing social inequalities between families. In the following, we present novel evidence for the case of Germany. We examined three dimensions of parental pressures: time pressure, financial pressure, and pressure that comes from the educational system. We investigated how the patterns of parental pressures differ by socioeconomic background and family form. In addition to distinguishing between lone parents and parents in couple families, we also paid special attention to the parenting pressures experienced by non-resident parents.

## 3. The German Context

### 3.1. Policy Context and Parental Time Use

In the international literature, Germany has commonly been classified as a conservative and familialistic welfare state that supports the single-earner model (Esping-Andersen 1999). The main policies that underpin this system are the option of joint taxation for married partners, the free coverage of the non-working spouse in the public health care system, and the tax exemption for marginal employment. Although these policies are still in place, family policies in Germany have undergone radical changes in recent years. In 2007, Germany introduced an earnings-related parental leave benefit, which was largely copied from the Swedish parental leave system (Leitner et al. 2008). Since 2005, the child care infrastructure in Germany has been significantly expanded. Unlike in countries such as the UK, which has a day care system that is mainly private, in Germany, child care is relatively inexpensive, and is mostly free of charge for low-income households. In 2013, a legal right to a public day care slot for all children aged one year or older was introduced, which further accelerated the already positive trend toward the use of public day care institutions. Figure 1 shows that only 8% of children under age three were in day care in 2006. By 2016, this share had risen sharply to 28%, which represents a 250% increase in a period of only 10 years. Partly as a legacy of the socialist era, child care has always been more widely available in eastern Germany (former German Democratic Republic) than in the western states of Germany. Against that background, the increase in child care usage has been less steep in eastern than in western Germany. For older preschool children (ages 3–5), the usage of public child care has become almost universal, and there have been no major changes in enrolment in the most recent years. However, many of the day care slots are used on a part-time basis only. While Germany has made tremendous advances in supporting the parents of children under age three, it offers less support to parents of older children. As many schools are still only part-time, parents often face considerable difficulties in arranging child care after their children enter primary school (Alt et al. 2019; Hüsken 2015).

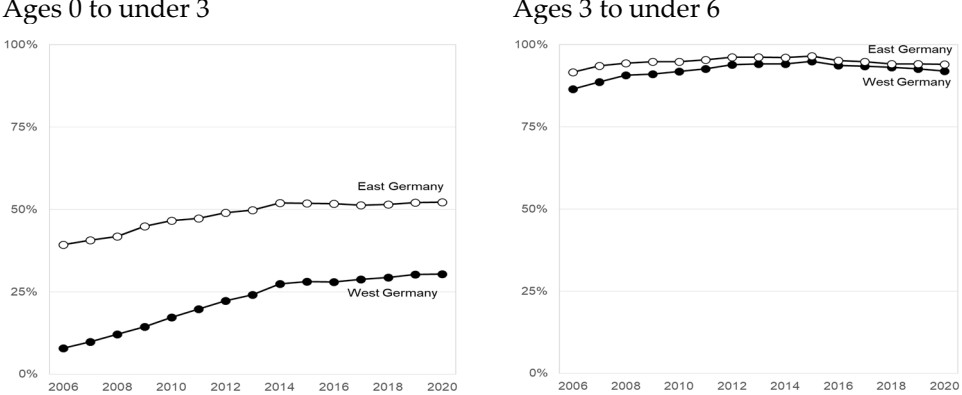

**Figure 1.** Usage of public day care by region and age of children. Source: Data compiled based on various publications of the German Statistical Office by akjstat. Note: These statistics include institutional child care as well as publicly funded child minders (Tagespflege).

The availability of day care shapes parental employment patterns. There is firm evidence that the 2007 reform led to an increase in the full-time employment rates of mothers, particularly in the second year after childbirth (Geyer et al. 2015). The parental leave regulations also included a quota that incentivized the mother and the father to split the leave between them: i.e., the parents would have to forfeit two months of leave if they did not share it. This "daddy quota" seems to have contributed to a rapid increase in the uptake of parental leave by fathers, even though many fathers took short leaves of two months only (Geisler and Kreyenfeld 2018).

Despite these major changes in family policies in recent years, employment patterns have remained strongly gendered in Germany. As Figure 2 (left panel) shows, in 1995–2019, fathers worked roughly 38 h per week, on average. While mothers' working hours increased over this period, mothers were still working only 17 h per week on average, or less than half the number of hours fathers were working. Correspondingly, mothers were spending substantially more time than fathers caring for their children (right panel of Figure 2). While mothers' time investments declined somewhat in the 2005–2009 period—which is the period when the abovementioned reforms were rolled out—fathers' time investments increased. Both fathers and mothers were spending more time with their children on the weekends (see also Samtleben 2019).

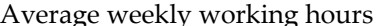

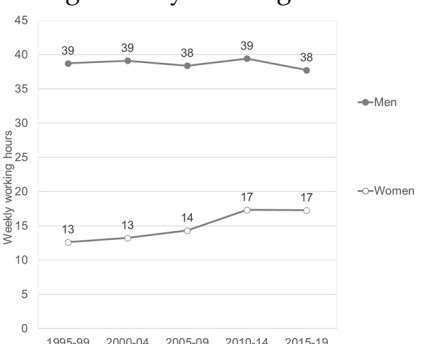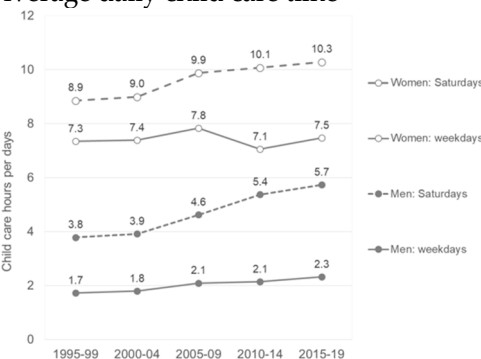

**Figure 2.** Average weekly working hours and average daily child care hours, women and men with children aged 12 and younger in the household, 1995–2019. Source: SOEP v36, own and weighted estimates (weighted by phrf). Note: Information for the time spent on child care on the weekends is only available every second year starting in 1995. The non-employed were included in the calculation of the mean working hours. Their working hours were set to zero.

Detailed time use data are collected for Germany every 10 years. A comparison of data from 2001/02 and 2012/13 showed that the time parents spent with their children increased over this period, particularly for fathers of children under age three (Klünder and Meier-Gräwe 2018; Meier-Gräwe and Klünder 2015; Walper and Lien 2018, p. 39). Most of the time (roughly 75%) fathers spent with their children was devoted to care activities and play. Compared to fathers, mothers spent more time on "routine tasks" (mothers: 50%, fathers: 36%) and less time on "interactive activities" (Walper and Lien 2018). Both fathers and mothers spent a significant fraction of time on transporting their child to day care, school, or child-related activities (BMFSFJ 2021). These types of time investments increased substantially over the study period. While the time parents spent on educational activities, such as on reading to their children and supporting them in their assignments (for school-aged children), made up only a small fraction of the overall time parents spent with their children, this share did increase slightly over the study period. The average number of minutes per day mothers spent on these activities increased from eight minutes in 2001/2 to 12 min in 2012/13 (for fathers, the corresponding increase was from two to four minutes per day) (BMFSFJ 2021, p. 155; Walper and Lien 2018, p. 39). Differences in the time spent on child care by family status were also reported: i.e., lone mothers spent less time with their children than partnered mothers, controlling for employment status

(Kahle 2004). No evidence exists for non-resident fathers, as this group cannot be identified with German time use data. The results of analyses of the time parents spend with their children by educational background have been mixed. Most studies for Germany have found no association (Berghammer 2013; Dotti Sani and Treas 2016; OECD 2017; Schulz and Engelhardt 2017). However, educational differences in the time spent reading to children have been found, with highly educated parents being especially likely to report that they read to their children (Schulz and Engelhardt 2017).

### 3.2. Prior Research on the Intensification of Parenting in Germany

Evidence on the "intensification of parenting" is still scarce for Germany. A mainly qualitative study that was commissioned by the Konrad Adenauer Foundation found that there are strong cleavages in parenting practices, with a dividing line running between active parents who consciously educate and intensively support their children, and parents who "let their children's development run its course" (Henry-Huthmacher et al. 2008, p. 8). The study also found that parenting practices are strongly influenced by social background, with parents of higher socioeconomic status being considerably more actively involved in childrearing than parents of lower socioeconomic status. These findings were also corroborated by data from the "Familienleitbild Survey" conducted in 2012. This study used an index of three items to operationalize "engaged parenthood."[1] The findings showed that parental engagement was more prevalent in the western than in the eastern states of Germany, and was strongly correlated with parents' education (Ruckdeschel 2015). In line with this finding, more recent evidence indicates that private schools are becoming more common in Germany, particularly in the east (Görlitz et al. 2018). Moreover, the probability that a child attends a private school was found to be strongly correlated with the parents' educational background.

Although these studies provided some insights into the role of socioeconomic determinants of family behavior, they paid little attention to the family context. The abovementioned time use research only reflected the patterns of parents who were living with their children. By contrast, there are no official data on the time use and parenting practices of non-residential parents.

### 3.3. Research Aims and Hypotheses

Our research adds to the sparse literature that exists on the "intensification of parenting" in Germany by analyzing parents' subjective parenting pressures. For that purpose, we used novel data from the study "Parenthood Today", which was conducted in 2019 in conjunction with the German Family Report. The data included item batteries that survey parents' subjective assessments of the pressures and the demands that they experience today. We focused on three areas: namely, (1) time pressure, (2) financial pressure reflecting rising demands for financial investments in children, and (3) pressure resulting from increasing demands on parental support for the education and promotion of their children. It is important to point out that time pressure referred to the pressure currently felt, whereas financial pressure and pressure from the educational system referred to changes across time.

Our data are cross-sectional, and thus focus only on subjectively perceived changes in pressures over time. Still, they complement more objective trend data in an important way. First, while these reported pressures are based on subjective assessments, they may have "real effects", as they can influence parental well-being and self-confidence. Secondly, examining "risk groups" and their subjective parenting pressures provides important insights for policy makers and counselors.

We examined two broad hypotheses. (1) We expected the majority of parents to experience increasing pressures in these three abovementioned domains (*intensification hypothesis*). (2) Assuming a widespread change of parenting norms across social groups as has been shown for the U.S. (Ishizuka 2019), we hypothesized that increasing pressures would be more strongly experienced by parents who lack the resources to follow this trend (*resource mismatch hypothesis*). For example, behavioral changes towards intensive

parenting have been shown to be more likely among highly educated parents (see Section 2). However, pressures of intensive parenting norms should be felt more strongly by parents with lower education, because they are less able to conform to these norms.

Apart from the differences by level of education, we also examined whether patterns differed by family form. Unlike prior research in this area, we included non-residential fathers in addition to two-parent families and single mothers. Perceived pressures were expected to be higher among single mothers than among mothers in couple households. We further hypothesized that the time and financial pressures of fathers would resemble those of mothers when controlling for employment status. Furthermore, we expected that the time spent working competes with the time available for child care, resulting in higher levels of time pressure among full-time employed parents.

## 4. Materials and Methods

### 4.1. Data and Analytical Sample

In the context of preparing the Ninth Family Report for Germany, the Institute for Demoscopy Allensbach was commissioned to conduct a nationwide study of parents of minor children in Germany. The objective of this survey was to gain insights into how parents experience parenthood today, and whether they believe that there have been any changes in the pressures, norms, and demands associated with "good parenting." For this purpose, personal interviews were conducted in October and November 2019 with a representative sample of German-speaking mothers and fathers with at least one child under 18 years of age in the household. In addition, a subsample of 160 interviews were conducted with separated fathers who were not (or were no longer) living with their children (two-stage quota selection). To make these data compatible with structural data from official statistics, they were factorially weighted (Institut für Demoskopie Allensbach 2020). The total survey sample consisted of 1688 respondents. For this investigation, we removed the small number of single fathers and non-resident mothers (as the sample size would have been too small to allow for a separate analysis of this group). After removing invalid information on the key variables of interest (see below), the analytical sample included 1652 cases. For further sample statistics, see Section 4.3 below.

### 4.2. Indicators

The main outcome variable used to measure *parents' time pressure* was a binary variable that distinguished between respondents who stated that they do not have enough time to engage with their children, and respondents who either reported that they have enough time, or said that they are undecided about this issue ("Taken everything together, do you have enough time for your child/ren? Or is there too little time?"). *Financial* pressure was measured with a binary variable based on a question that asked respondents whether they believe that raising children has become more expensive ("It is more expensive to have children nowadays than it was in the past."). The *pressure from the educational system* was measured using a binary variable that indicated whether the respondents believe that the demands of the educational system have increased ("The demands on children's education and support have increased significantly."). Note that the two latter questions have a retrospective element, as they prompt the respondent to compare their current situation to the past. However, as the "past" is not further specified, it is unclear whether the respondents were comparing their parental situation with that of their own parents, or with their own experiences in earlier times.

One key independent variable of interest was the family form. We distinguished between (1) men in couple households, (2) women in couple households, (3) lone mothers, and (4) non-residential fathers. Non-residential fathers were defined as men who had children (under age 18) from a prior partnership, but who were not sharing a household with them. These men may, however, have been living with a new partner. Lone mothers were women who were sharing a household with their child or children (under age 18), but not with a partner. The men and women living in couple households were parents

who were sharing a household with their partner and their child or children (under age 18). Note that we did not differentiate between nuclear and stepfamily couple households or married and cohabiting families. While considering these families separately would have provided important additional dimensions of family diversity, the numbers of stepfamilies and cohabiting families were too small to allow us to do so. Same-sex couples with children, who also would have provided an additional important dimension of family diversity, were not part of this study. Furthermore, lone fathers and non-resident mothers were not included in our sample due to small sample sizes (see above).

The socioeconomic background was operationalized over the level of education, distinguishing between low ("Hauptschule" or less), medium ("Realschule" or equivalent), and high ("Abitur" or "Fachhochschulabschluss") levels of school education. The employment status distinguished between full-time employed (35 h or more), part-time employed (less than 35 h), and not working at the time of the interview. Age of the youngest children was included in the models as a categorical covariate (ages 0–2, 3–5, 6–11, 12–17 years). We also took into account the number of children of the respondent (one, two, three or more). Given the regional variations in family forms and maternal employment, we also considered whether a respondent was living in the eastern or western part of the country.

### 4.3. Sample Statistics

Table 1 reports the sample statistics broken down by family form (fathers and mothers in couple households, single mothers, and non-resident fathers). The table shows that non-resident fathers and lone motherhood were more prevalent in eastern than in western Germany. Lone mothers, and to a lesser extent also non-resident fathers, were more likely to have only one child than fathers and mothers in couple households. As expected, the age of the youngest child was lowest in the couple households. We also observed a pronounced educational gradient. Lone mothers had lower educational levels than mothers in couple households. The differences between men in couple households and non-residents fathers were less pronounced. However, a larger fraction (52%) of the fathers in couple households than of the non-resident fathers (46%) had a high school degree. We also found large differences by gender in employment behavior. Only 25% of women, compared to 91% of men, in couple households worked full-time. Moreover, 87% of non-resident fathers were full-time employed. Lone mothers were more likely to be full-time employed (44%) than mothers in couple households but were much less likely to be full-time employed than non-resident fathers.

**Table 1.** Sample statistics by family form (in column %).

|  | Couple Households | | Non-Resident | Lone |
|  | Fathers | Mothers | Fathers | Mothers |
|---|---|---|---|---|
| **Region** |  |  |  |  |
| Western Germany | 82 | 78 | 77 | 73 |
| Eastern Germany | 18 | 22 | 23 | 27 |
| **Number of children** |  |  |  |  |
| One child | 43 | 40 | 45 | 65 |
| Two children | 43 | 42 | 40 | 25 |
| Three children | 14 | 18 | 15 | 9 |
| **Age of youngest child** |  |  |  |  |
| Age 0–2 | 29 | 27 | 6 | 13 |
| Age 3–5 | 20 | 20 | 14 | 20 |
| Age 6–11 | 29 | 30 | 39 | 29 |
| Age 12–17 | 23 | 23 | 41 | 39 |
| **Education** |  |  |  |  |
| Low | 15 | 15 | 16 | 22 |
| Medium | 33 | 38 | 38 | 44 |
| High | 52 | 47 | 46 | 34 |
| **Employment** |  |  |  |  |
| Full-time | 91 | 25 | 87 | 44 |
| Part-time | 6 | 50 | 6 | 41 |
| Not employed | 3 | 25 | 8 | 15 |
| N | 624 | 707 | 161 | 160 |

## 5. Results

### 5.1. Descriptive Findings

Table 2 reports descriptive findings for the three outcome variables. The share of parents who said they feel time pressure (average of 67%) was similar to the share of parents who reported experiencing increased financial pressure (average of 64%) and educational pressure (average of 69%). Although the averages were similar, there were substantial variations across family forms and parental educational levels.

Regarding time pressure, the results showed that women in couple households rarely reported that they do not have enough for their children (26%), whereas men in couple households more often reported experiencing time conflicts (45%). Moreover, slight majorities of both lone mothers and non-resident fathers (both 52%) indicated that they feel that they do not devote enough time to their children. While there were large and significant differences between family forms (tested by a chi-square independence test), parents' education did not seem to be associated with perceptions of time conflicts.

Perceptions of financial pressure were found to vary significantly by educational level (tested by a chi-square independence test). As expected, parents with low levels of education were more likely to report that their financial pressures had increased. Regarding variations by family form, differences between single mothers and mothers in couple households proved significant (test of equivalence of proportions).

Regarding the pressure from the educational system, parents with low educational resources said they feel slightly more pressure from the educational system than parents with medium or high education. Rather surprisingly, we found that women in couple households expressed more concern about the demands of the educational system than did parents in other family forms. Indeed, mothers in couple households reported worrying much more about pressure from the educational system than about time or financial pressures. For separated parents, financial concerns trumped concerns about pressure from the educational system.

**Table 2.** Concerns about having too little time for children (TIME), concerns that financial requirements have increased (MONEY), concerns that educational requirements have increased (EDUCATION), in %.

|  | Time | Money | Education |
|---|---|---|---|
| **Family form** |  |  |  |
| Man: Couple household | 45 | 63 | 65 |
| Woman: Couple household | 26 | 64 | 72 |
| Woman: Lone mother | 52 | 73 | 69 |
| Man: Non-resident father | 52 | 67 | 65 |
| **Education** |  |  |  |
| Low | 34 | 72 | 72 |
| Medium | 38 | 68 | 68 |
| High | 36 | 57 | 68 |
| Total | 67 | 64 | 69 |

Note: Weighted estimates.

### 5.2. Multiple Regressions

#### 5.2.1. Analytical Strategy

In the following, we present results from logistic regressions that model the determinants of intensified parenthood. As above, we distinguished between the three types of pressure associated with having children, namely the time pressure, the financial pressure, and the pressure from the educational system (TIME, MONEY, EDUCATION). Our main independent variables of interest were the family form and the educational background of the parent. We controlled for region (eastern/western Germany), the child's age, and the number of children of the respondent, as well as for the respondent's employment status. Findings are reported as average marginal effects (AME). The descriptive statistics revealed large differences in time pressure by family form. It is likely that these differences may be

moderated by employment status. For example, one could assume that full-time employed lone mothers are under particularly heavy time pressure as they do not have a partner who can take over child care tasks. For that reason, we also included the interaction of employment status and family form. This allows us to check whether the patterns within the group of employed parents were uniform across family forms. We visualized these AME in a figure to facilitate the interpretation of the results.

### 5.2.2. Determinants of Parental Concerns

Table 3 reports the results from the logistic regression models. The column labelled "TIME" reports the results with the time demands as outcome variables. The model results confirmed the large differences by family form that were already reported in the descriptive statistics. The model also showed that subjective time pressures were significantly more pronounced in eastern than in western Germany. Surprisingly, the age of the youngest child and the number of children in a household were not found to be associated with parents' feelings that they do not have enough time for their children. Parental education was also shown to have no significant effects on the outcome variable. However, employment was found to be a powerful predictor of the likelihood of parents reporting concerns about spending too little time with their children. For example, the probability of indicating such concerns differed by 41% between not employed and full-time employed parents.

**Table 3.** Logistic regression model, dependent variable: 1: concerns, 0: no concerns. Average predicted probabilities (AME).

|  | Time | Money | Education |
|---|---|---|---|
| *Family type and gender* | | | |
| Man: Couple household | 0.056 | 0.001 | −0.063 *** |
| Woman: Couple household | Ref. | Ref. | Ref. |
| Woman: Lone mother | 0.190 *** | 0.058 | −0.036 |
| Man: Non-resident father | 0.094 ** | 0.022 | −0.078 ** |
| *Region* | | | |
| Western Germany | Ref. | Ref. | Ref. |
| Eastern Germany | 0.057 ** | 0.049 * | −0.076 *** |
| *Age of youngest child* | | | |
| Age 0–2 | Ref. | Ref. | Ref. |
| Age 3–5 | 0.057 | 0.001 | 0.025 |
| Age 6–11 | 0.017 | −0.001 | 0.068 * |
| Age 12–17 | −0.016 | 0.029 | 0.010 |
| *Number of children* | | | |
| One child | Ref. | Ref. | Ref. |
| Two children | −0.008 | −0.009 | 0.029 |
| Three children | −0.009 | −0.036 | −0.059 * |
| *Education* | | | |
| Low | Ref. | Ref. | Ref. |
| Medium | 0.011 | −0.070 ** | −0.022 |
| High | 0.013 | −0.154 *** | −0.036 |
| *Employment* | | | |
| Full-time | Ref. | Ref. | Ref. |
| Part-time | −0.228 *** | −0.021 | −0.009 |
| Not employed | −0.411 *** | −0.010 | −0.007 |

Note: * $p < 0.1$; ** $p < 0.05$; *** $p < 0.01$.

Findings on parents' perception that raising children has become more expensive are reported in the column "MONEY". While the descriptive statistics suggested that lone mothers were most likely to experience an increase in child costs, the multiple regression showed that this difference in financial pressure by family form was no longer significant when accounting for socioeconomic resources, which differ by family form. Like in most other countries, lone mothers in Germany are less likely than partnered mothers to have a high educational degree (see also Table 1). These background variables seem to be important for understanding lone mothers' financial concerns about increasing child costs. The table

shows that compared to the reference group of low educated parents, the probability of experiencing increasing child costs was 7% lower for the medium educated parents and was 15% lower for the highly educated parents. The model also found that eastern German parents were more likely to be concerned over a rise in children's costs than parents in western Germany. The other covariates did not have a significant effect on the outcome variable.

The third model (column "EDUCATION") analyzed the pressure that comes from the educational system. In line with the prior descriptive findings, the regression model supported that women in couple households were more likely to feel education-related pressure than resident and non-resident fathers. Respondents in eastern Germany, as well as parents with three or more children (compared to parents with only one child), proved less concerned about education-related pressure, possibly because other concerns, such as financial worries, crowded out their concerns about education. The child's age also mattered. Parents with primary school-aged children (ages 6–11) were more concerned than parents with younger and older children. As children's performance in primary school grades determines which school track they are assigned to at the secondary level in Germany, this phase seems to be particularly stressful for parents.

In a final step, we estimated an interaction model of employment status and family form. We did so to examine whether certain constellations, such as lone and full-time employed mothers, reported being under additional time pressure. As the sample sizes of part-time employed and non-working fathers were rather small, we had to group part-time and non-working into one category. Figure 3 reports the average predicted probabilities (AME) from this investigation. The figure clearly shows that the full-time employed parents were the most likely to report not having enough time to support their children than other parents (part-time and not employed parents). Of these parents, lone mothers, but also non-resident fathers, were especially likely to express concerns. Roughly 70% of the full-time employed lone mothers and 60% of the full-time employed non-resident fathers said they feel they cannot devote enough time to their children. By contrast, only about 20% of non-working or part-time employed mothers in couple households reported that they worry that they are spending too little time with their children. Although the interaction model is insightful, it should be noted that according to the likelihood ratio test the interaction did not improve the fit of the model significantly.

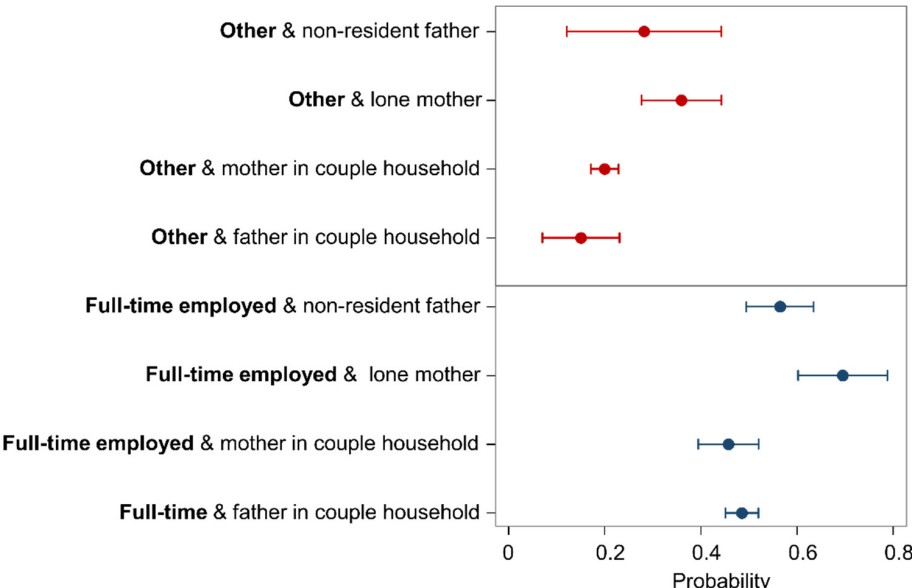

**Figure 3.** Average predicted probabilities from logistic regression model and 95%-confidence bounds, dependent variable: 1: concerns, 0: no concerns. Note: Effects were controlled for parental education, number of children, age of children, region.

## 6. Summary and Conclusions

### 6.1. Discussion of Findings

This paper has contributed to the literature on the intensification of parenting by providing novel evidence based on data from the "Parenthood Today" survey, which was conducted in 2019 in conjunction with the German Family Report (BMFSFJ 2021). Focusing on parents' perceptions of pressures which result from rising child costs, increasing demands to support children's education, and appropriate time investments in children, we expected that the majority of parents would experience increasing pressures in these domains of parenting (*intensification hypothesis*). Furthermore, we hypothesized that parents with fewer resources would be more likely to experience an increase in pressures than more advantaged parents (*resource mismatch hypothesis*). We examined differences by parental education, but also paid special attention to differences by family form. In contrast to prior research in this area, our analyses included non-resident parents in addition to parents in couple households and lone parents. The "Parenthood Today" study was uniquely suitable for this purpose, as it included an oversample of non-resident fathers.

In line with the intensification hypothesis, the overwhelming majority of parents said they feel that the pressures they are experiencing are greater today than they were in the past. However, regarding the resource mismatch hypothesis, we did not find consistent patterns by socioeconomic background. Whereas low educated parents, compared to parents with higher education, reported experiencing a greater increase in financial pressure due to increasing child costs, they did not say that they feel more time pressure or more pressure from the educational system. Hence, these findings are not in line with the resource mismatch hypothesis. Overall, our results did not align with previous findings from the Anglo-American literature, which showed that the pressures associated with the labor market and the educational system translated into marked differences in parents' perceptions and behavior depending on their socioeconomic background. One of our findings that may be seen as alarming was that parents with children of primary school age were more likely than other parents to report feeling pressure from the educational system. A particular feature of the German system is that educational tracking happens early, toward the end of primary school (typically in grade 4, when children are around age 10). Our results suggested that this tracking may place pressure on parents.

A clear finding from our investigation was that the time pressures reported by parents differed substantially by family form. Lone mothers, but also non-resident fathers, expressed considerable concerns that they are unable to spend sufficient time with their children. When we distinguished subgroups based on family form and parents' working hours, we found that full-time employed lone mothers and non-resident fathers were the groups who reported experiencing the most acute time pressure. With the expansion of public child care, the conditions for combining work and family life have greatly improved in Germany. At the same time, female employment rates have increased. Unlike in the Anglo-American countries, where the employment rates of lone mothers often lag behind those of partnered mothers, the patterns are reversed in Germany. Compared to partnered mothers, lone mothers were more likely to work full-time, which is likely to reduce their time available for childcare.

However, non-residential fathers also reported experiencing greater pressures than resident fathers, even though most of the men in both groups were working full-time. This finding may be seen in the context of the legal barriers to shared physical custody in Germany, which have made this custody arrangement relatively rare in Germany (Walper et al. 2021a). Yet we should also note that the self-image of separated fathers in Germany has been changing rapidly, as these fathers are increasingly demanding a more solid and flexible legal basis for remaining involved in the lives of their children after separation. Currently, various proposals to reform physical custody so that non-resident parents can spend more time with their children are under review (Walper et al. 2021b). While engaged fatherhood is on the rise, it is very uncommon for fathers to reduce their working hours to take care of their

children, regardless of whether they are separated or in a union. This may result in fathers experiencing greater tension between their role as a provider and as a carer.

### 6.2. Limitations

Our study has highlighted the importance of integrating non-resident parents into the debate on the intensification of parenthood. However, it also had several limitations. Most importantly, our analysis was cross-sectional only and relied on subjective perceptions of change. Respondents were asked to evaluate whether they feel that that the pressures they are currently experiencing are greater than in the past. This is obviously a very subjective way of assessing time trends in parenting. Nevertheless, the high prevalence of increases in pressures perceived by parents are in line with more objective data and reflect important aspects of parents' experiences in the parenting role. Another limitation of our investigation was that we focused on three broad categories of parental investments (namely: time, money, education). Such a broad approach provided a good overview of different dimensions of parenting, but it does not account for the many nuanced ways that parents invest in their children. For example, although parents with low education experienced similar time pressure to highly educated parents and may have spent the same amount of time with their children, other research has shown that highly educated parents are more likely to use the available time to involve their children in stimulating educational activities (e.g., Davis-Kean et al. 2021; Dermott and Pomati 2016).

More detailed data, such as time use surveys, are better suited than our data to unravel parents' multi-faceted time investments in their children. German time use surveys are collected every 10 years. Sadly, it is not possible to identify non-residential parents in the German data. Thus, we cannot measure the time investments and time pressures of non-resident fathers using these data. We hope that our analysis inspires efforts to collect better data in future rounds of time use surveys that include not only detailed information on time use, but also appropriate measures of family diversity.

### 6.3. Implications for Parenting and Parent–Child Relationships in the 21st Century

The wide spread of intensive parenting norms bears chances for children, whose wellbeing and development may benefit from increased parental investments. However, it also bears risks, not only for children, but even more so for parents. If parental involvement is not adjusted to children's growing competencies and needs for autonomy, children's self-reliance and development of personal responsibility may suffer (e.g., Perry et al. 2018). For parents, tight social norms of involved parenting combined with difficulties to meet these demands are likely to contribute to increasing parental pressures, as supported by our analysis. Pressures to be perfect in the parenting role have been linked to elevated levels of parental stress and compromised feelings of competence and wellbeing (Meeussen and Van Laar 2018). Counselors face the challenge of balancing two competing goals: on the one hand the goal to promote parents' skills and competencies in child rearing, and, on the other hand, the goal to shield parents from stress, which may compromise their self-confidence. Our findings suggest that these pressures are not limited to parents with fewer resources. In fact, time pressure and the pressure to promote children's education were similarly felt by parents of all educational groups. Also, among couples, pressures were not more pronounced for mothers than for fathers. This may indicate that the intensification of parenting also concerns fathers (including non-residents ones) who should be better addressed and involved in parenting support and counseling.

**Author Contributions:** Conceptualization, S.W.; Formal analysis, M.K.; Supervision, S.W.; Writing—original draft, S.W. and M.K. All authors have read and agreed to the published version of the manuscript.

**Funding:** This research received no external funding.

**Institutional Review Board Statement:** Ethical review was not required for the survey "Elternschaft heute" because no personal information were stored.

**Informed Consent Statement:** Informed consent was obtained from all respondents of the survey "Elternschaft heute" by the Survey Agency (Institute for Demoscopy Allensbach).

**Data Availability Statement:** The data of the survey "Elternschaft heute" is available for re-analysis upon request from the survey agency (Institute for Demoscopy Allensbach; Email: whaumann@ifd-allensbach.de). Stata-code is available upon request from the authors.

**Conflicts of Interest:** The authors declare no conflict of interest.

## Note

[1] The items were: (1) "Parents should completely put aside their own needs for their children"; (2) "Children grow up anyway, you don't have to worry so much"; and (3) "Parents can do a lot of things wrong when it comes to parenting, so they need to inform themselves well".

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
