# Peer review of "The Intensification of Parenting in Germany: The Role of Socioeconomic Background and Family Form"

_socsci, doi:10.3390/socsci11030134_

Round 1

Reviewer 1 Report

The authors report findings from a large-scale cross-sectional survey study of parents in Germany. They focus on three key responses from the parents – how they report subjectively time, financial and educational pressures.

The study appears to have been well-executed overall, and the authors have uncovered some interesting details about these various pressures, but I find the results are quite tenuously linked to the literature on ‘intensive parenting’. The data is so thin that these claims are very difficult to make. As I finish the article, I find I have learned:

- that poorer parents are more likely to report financial pressure;

- time poor parents, including fathers that live separately from their children, report feeling time pressure,

- and then primary-school parents worry more about the educational outcomes of their children, which is very well explained by the authors in terms of system of education in Germany.

That is, the findings uncovered are hardly novel or unexpected. The first half of the article discusses in quite a lot of depth the intensive parenting literature, but I think this article as it stands does not actually add much to it. How do these ‘pressures’ actually relate to intensive parenting (if at all)? A more direct link needs to be made with some clear hypothesis (though I am not sure it is possible with these questions/data). For example, the authors seem to suggest that the parents are expressing a pressure to spend more time with their children, but actually it looks like they are expressing an actual deficit of time. Therefore, with the results and data as they stand, I do not think this claim can be substantiated:

Full-time employed lone mothers, but also non-resident fathers, feel severe pressure to accommo- 12 date their role as workers and carers. Our results draw attention to the need to better integrate the 13 needs of post-separation families, including non-resident fathers, in the debate on the “intensifica- 14 tion of parenting”.

 Moreover, as pointed out by the authors, the subjective idea that ‘things have changed’ or gotten worse as reported in their binary responses (which was linked with the intensive parenting narrative that pressures etc have increased), does not actually tell us much.

For a more robust and interesting article, I suggest that authors return to a sub-sample of the participants for more in-depth research.

Alternatively, or additionally, one might spend much more time considering how structural constraints are shaping parents’ perceptions and pressures. Either way, the current claims cannot be upheld.

Some smaller notes:  

Abstract: ‘Increased pressure’ compared to what? (we later find there is no definitive point, so subjective perception of change should be emphasised in the abstract)

How does nursery care etc relate to intensive parenting? You may like to compare with other euro-american countries where subsidised childcare barely exists (eg UK has the most expensive childcare in Europe…)

Author Response

We are very grateful for your critical reading of our manuscript and your valuable suggestions of how to improve it. All issues raised are addressed in the attached pdf-file. We start with our response to the editor who pointed out the salient issues raised in the reviews and then follow with our response to your review. Some of your points are addressed in more detail in our response to the editor.

Reviewer 2 Report

This is a solid paper - measured by the standards of this journal. It presents novel German data on parenting. It specifically distinguishes time, money, and education pressures as perceived by parents. It highlights differences by parental SES and household structure. It documents that - unlike many anglosaxon countries- education has little effect on time and educational pressures.

This paper would, however, benefit from a more detailed conceptualization of the key concepts, especially intensification of parenting. It appears to have both a behavioral and evaluative element. I have a related doubt about the measurement- subjective assessment of current costs and demands (presently, children are "more expensive" or "more demanding" to raise than in some unspecified older period) appears to tap only one dimension of the concept.

Time pressure seems to be related (conceptually) only to changing expectations and norms, but it clearly also related to the kinds of jobs people have. It is suprising that the paper makes so little reference to the sociological literature on role conflict and work-family reconciliation. Socio-economic status should be more directly operactionalized, education is a less adequate measure than class/occupational status.

A broader discussion of ongoing societal changes (in labor markets, institutions, values etc.) could clarify what developemtns we can expect and how to think about cross-country differences. Most importantly, the paper assumes that children and values are exogenous and people have to accomodate. Yet, children can also be endoganmous to values and adults can accomodate prevailing values by having fewer (or no) children and thus reduce presures in this manner. This option is entirely missing from the literature review.

Modelling: I would prefer that the reference category would be the same across models. Unreported AME are a more adequate instrument for model comparison than odds ratios.

Author Response

(The authors gave the same response as above.)

Round 2

Reviewer 1 Report

The article has improved significantly after the revisions. Well done for your extensive work. 

Some minor points: 

  • I think the distinction between the three items reported on (time, money, education) is sometimes lost. The time measurement, does not measure an increase in time, rather perceived lack of time. The other two measurements offer a subjective perception of increased pressures around education and money. The wording should be amended to maintain this clarity and some reflection on how the differences between these three may or may not influence the findings. For example, in the abstract the authors right 'In each of these domains, more than 60% of the parents experienced increased pressure.' Whereas, in actual fact, in the time dimension, they reported not having enough time (not that it had altered over time) and in the other two, they reported a perception that pressures had increased.
  • The second paragraph introduces arguments which have not yet been explained. I think some small editing should be made here, to show that the authors will make these arguments in their article. 
  • Be careful about slippage around the use of 'European' as contrasted with the UK. The UK is still in Europe, even if not in the EU. (Moreoever, Ireland has a similar legal and policy context to the UK, but is both European and in the EU.)
  • Page 7, final paragraph. Stats are compared around work hours. Perhaps better to use paid and unpaid work here, or another descriptor, since this seems to suggest that domestic labour (caring or otherwise) is not work. 
  • P17, line 699-701, the authors state 'Compared to partnered mothers, lone mothers were more likely to work full-time, which likely made it harder for them to devote sufficient time to their children.' I think this suggests a value judgement on what is or is not 'sufficient time' with children. I suggest changing to 'Compared to partnered mothers, more lone mothers worked full-time, which is likely to reduce their time available for childcare.'

Author Response

Thank you very much for your careful review of the revised version of our manuscript. We have taken up all suggestions and responded to the issues as explained in the attachment.
